# Causal Imputation via Synthetic Interventions

**Chandler Squires**[*]                                        CSQUIRES@MIT.EDU
*LIDS, IDSS, and CSAIL, MIT, Cambridge, MA, USA*

**Dennis Shen**[*]                                          DSHEN24@BERKELEY.EDU
*LIDS, MIT, Cambridge, MA, USA*

**Anish Agarwal**                                    ANISH.AGARWAL@GMAIL.COM
*LIDS, MIT, Cambridge, MA, USA*

**Devavrat Shah**                                          DEVAVRAT@MIT.EDU
*LIDS, MIT, Cambridge, MA, USA*

**Caroline Uhler**                                            CUHLER@MIT.EDU
*LIDS and IDSS, MIT, and Broad Institute, Cambridge, MA, USA*

**Editors:** Bernhard Schölkopf, Caroline Uhler and Kun Zhang

## Abstract

Consider the problem of determining the effect of a compound on a specific cell type. To answer this question, researchers traditionally need to run an experiment applying the drug of interest to that cell type. This approach is not scalable: given a large number of different actions (compounds) and a large number of different contexts (cell types), it is infeasible to run an experiment for every action-context pair. In such cases, one would ideally like to predict the outcome for every pair while only needing outcome data for a small *subset* of pairs. This task, which we label *causal imputation*, is a generalization of the causal transportability problem. To address this challenge, we extend the recently introduced *synthetic interventions* (SI) estimator to handle more general data sparsity patterns. We prove that, under a latent factor model, our estimator provides valid estimates for the causal imputation task. We motivate this model by establishing a connection to the linear structural causal model literature. Finally, we consider the prominent CMAP dataset in predicting the effects of compounds on gene expression across cell types. We find that our estimator outperforms standard baselines, thus confirming its utility in biological applications.

**Keywords:** Latent factor model, imputation, causal inference

## 1. Introduction

A central goal in science is to determine the outcome of an action (i.e., intervention). Traditionally, researchers rely on experimentation to answer such questions. However, this approach is rarely scalable and/or can be unethical in many critical applications, e.g., clinical trials and policy evaluation. As a result, there is a growing interest in utilizing already-existing data to help predict the effect of every candidate action, prior to deciding which action to execute. The key challenge underpinning this approach is that the available data regarding the effect of an action almost always comes from a *different context* than the one in which we are interested in making a prediction. To further motivate our problem setting, we provide an example from healthcare below.

*A Motivating Example.* Consider a prevalent medical scenario in which physicians aim to re-purpose existing drugs to treat novel diseases such as COVID-19. Although there are over 20,000

---
[*]Equal contribution

small molecule compounds (actions) with known therapeutic capabilities, exposing COVID-19 infected cells (new contexts) to each compound would be far too time-consuming and costly. Fortunately, the publicly available CMAP dataset (Subramanian et al., 2017) has already catalogued the effects of these compounds on a variety of other cell types (different contexts). In such a setting, the primary intellectual challenge is to develop a methodology which uses the CMAP dataset to predict the outcome of applying each compound on COVID-19 infected cells. Such a method leads to efficient *hypothesis generation* for finding compounds with therapeutic potential for this disease.

*Key Question.* The focus of this work is to tackle problems such as the one described above. More generally, we aim to answer the following question:

> *"Given a dataset of outcomes from various action-context pairs, can we predict the outcome of a given action on a context in which it has not been applied?"*

**Our Contributions.** The main contributions of this work are three-fold: algorithmic, theoretical, and empirical. We summarize them below.

*(i) Algorithmic.* We extend the recently proposed synthetic interventions (SI) estimator (Agarwal et al., 2020) to handle more general sparsity patterns. At its core, the SI estimator produces a counterfactual prediction for any "target" (novel) context-action pair by first constructing a "synthetic" version of the target context as a weighted combination of "donor" (different) contexts via (regularized) regression. Then, it uses the learnt model to re-scale the outcomes of those donor contexts under the target action to estimate the outcomes of the target context-action pair. Taking inspiration from biological applications, we adapt the SI estimator to instead construct a synthetic version of the target *action* as a weighted combination of donor actions. Given the more general observation patterns, we also equip the SI estimator to use *all* available data when learning the regression model. As a result, the SI estimator of Agarwal et al. (2020) becomes a special case of our formulation. Moving forward, we refer to our estimator as `SI-A`. For details, please refer to Section 4.

*(ii) Theoretical.* We justify our algorithmic approach by presenting an "identification" result in Theorem 1. More formally, we establish that our estimator yields the correct causal estimates for the causal imputation task, under a factor model across contexts and actions. Further, we re-interpret the SI causal framework by connecting it to linear structural causal models (see Proposition 1).

*(iii) Empirical.* Using the prominent CMAP dataset, we extensively benchmark our approach against well-established baselines. The CMAP dataset contains gene expression signatures for over 20,000 different small molecule compounds (actions) across 70 different cell types (contexts). Using a small fraction of the observed signatures as a training set, we consider the task of imputing the signatures associated with a held-out test set that emulates an un-experimented collection of novel cell type-compound pairs. Empirically, we find that our estimator outperforms the baselines. In particular, the median normalized root mean-square error (NRMSE) of `SI-A` is 0.34, compared to 0.41 for the closest baseline. Similarly, `SI-A` improves the *alignment* of the predicted effect of the action with the true effect of the action, measured in terms of cosine similarity, improving from 0.44 in the best baseline to 0.68. Moreover, our experiments show that our variant of the SI estimator (which regresses along compounds) significantly outperforms the original SI estimator (which regresses along cell types), a result that may be of independent biological interest.

**Organization of the paper.** In Section 2, we review related work on causal imputation. Then, in Section 3, we formally state our problem setting. In Section 4, we describe our estimation strategy, which extends the original SI estimator. We state our formal results in Section 5, establishing an

| | Fast | Heterogeneous Effects | Guarantees | Needs selection diagram |
|---|---|---|---|---|
| Fixed Effects (FE) | ✓ | ✗ | ✓ | ✗ |
| Autoencoding + FE | ✓ | ✓ | ✗ | ✗ |
| Causal Transportability | ✓ | ✓ | ✓ | ✓ |
| MICE/MissForest | ✗ | ✓ | ✗ | ✗ |
| Synthetic Interventions | ✓ | ✓ | ✓ | ✗ |

Table 1: Summary of Causal Imputation Methods

identification result under a factor model assumption, as well as providing an interpretation of this assumption in terms of a linear structural causal model. Finally, Section 6 showcases our empirical findings on the CMAP dataset.

## 2. Related Work

Predicting the (vector-valued) outcome of an action in a novel context, a task we call *causal imputation*, is a ubiquitous problem, and there are numerous lines of study tackling this challenge. We highlight some of the most relevant below. In practice, we require a method that (i) is *fast*, (ii) can model *heterogenous* effects across different cell types, (iii) comes with theoretical *guarantees*, and (iv) does *not* need a detailed model of the causal system as input. Existing methods, which we describe below, all lack one of these 4 desiderata, as summarized in Table 1.

**Fixed effects.** The simplest model for the effect of actions across different contexts is the *fixed effects* model, which associates each action with a vector representing the additive effect of that action, assumed to be *invariant* across contexts. We note that the fixed effects estimator is a special case of the regularized linear model proposed in Dixit et al. (2016). Their method makes predictions of the form $\widehat{\mathbf{x}}^{ca} = \beta^a + \mathbf{w}^c \beta^c$, where $\widehat{\mathbf{x}}^{ca}$ denotes the *outcome* vector associated with cell type $c$ under action $a$ and $\mathbf{w}^c$ is a set of covariates for the cell type $c$. When no covariates are available except a one-hot encoding of the cell type, their estimator reduces to the fixed-effects model $\widehat{\mathbf{x}}^{ca} = \beta^a + \beta^c$. Under a fixed effects model, the effect of an action in any new context can be predicted based solely on the effect of that action in a *single* other context, greatly reducing the number of required experiments. However, the fixed effects model is not realistic in our application of interest, as it does *not* allow for heterogeneity, even though the effect of a drug on the expression of a given gene can vary drastically between cell types (Kidd et al., 2016), e.g., increasing the expression of a gene in one cell type while decreasing it in a different cell type.

To overcome this limitation of the fixed effects model, Lotfollahi et al. (2019) introduced the *scGen* method. This method first trains a variational autoencoder (VAE) in order to find a low-dimensional representation (latent embedding) of gene expression data, then uses a fixed effects model in the *latent* space of the VAE. Similarly, one could instead apply our method, SI-A, in the *latent* space. The theoretical study of combining autoencoders, or other nonlinear embeddings, with "direct" causal imputation methods such as the one considered here, is an interesting direction for future work. To the best of our knowledge, no theoretical guarantees yet exist for such combinations, but promisingly, recent work using autoencoders to predict perturbation effects on SARS-CoV2 infected cells empirically demonstrated that autoencoders *align* representations in a way that induces linearity amongst the perturbations and cell types in the learned space (Belyaeva et al., 2020).

**Causal transport.** The goal in causal transport is to find a *transport formula* (Bareinboim and Pearl, 2014, 2016), i.e., a map from the outcomes of actions in a set of "source" contexts to the outcome in a "target" context under some action. Whereas causal transport focuses on prediction from data generated in *different* contexts than the target context, causal imputation generalizes this problem to *also* use data about *different* actions, including from the target context itself. Previous methods algorithmically derive transport formulas from a *selection diagram*, i.e., a causal model of the system that introduces additional nodes corresponding to contexts, allowing one to model how causal mechanisms might change between contexts. See Lee et al. (2020b) for a recent exposition on selection diagrams and algorithms for deriving transport formulas. A major contrast between the present work on causal imputation and prior work on causal transport is that SI-A does *not* rely on specifying (or learning) a selection diagram. Indeed, as we show in Proposition 1 and Theorem 1, SI-A is consistent for *any* linear causal structural model within a large class. A further advantage of the factor model framework considered in this work is that the key assumption (Assumption 2) that enables SI-A to generalize from a small collection of experiments to predicting on untested context-action pairs can be tested via a simple data-driven hypothesis test (see Section 4).

**Traditional imputation methods.** As missing values are a common issue in many applications (Bell and Koren, 2007; Troyanskaya et al., 2001), a number of imputation methods have been developed to fill in missing entries, see e.g. Bertsimas et al. (2017) for a recent review. Two prominent methods which are widely used in imputing genomic and other biological data (Tan et al., 2017; Waljee et al., 2013) are MICE (Van Buuren and Oudshoorn, 1999) and MissForest (Stekhoven and Bühlmann, 2012). To the best of our knowledge, these methods have no formal identification guarantees. Moreover, they are several orders of magnitude slower than our SI-A estimator - see Appendix F.1 for a comparison to MICE - making them impractical for the problem considered here.

The linear factor model assumption (Definition 1) used by our model is similar to factor model assumptions made by methods for low-rank tensor decomposition, such as PARAFAC Harshman et al. (1970); Bro (1997). As in the matrix case (Chatterjee, 2015; Agarwal et al., 2018; Sportisse et al., 2020), such decompositions can be useful tools for tensor estimation (Jain and Oh, 2014), especially in the missing-at-random (MAR) or missing-completely-at-random (MCAR) settings (Rubin, 1976). However, PARAFAC assumes a low-rank *CP decomposition*, which is more restrictive than the low-rank *mode-2 decomposition* in Definition 1. Furthermore, PARAFAC requires a costly alternating minimization scheme to optimize, so that it is again several orders of magnitude slower than our SI-A estimator.

**Methods using other data modalities.** The imputation methods we have thus far discussed all begin with genomic data from various compounds and cell types, and impute genomic data for *novel* pairs of compounds and cell types. Orthogonally, there is a vast literature on predicting perturbation response using *different* data modalities, e.g. images (Hofmarcher et al., 2019; Yang et al., 2018) or molecular structure (Stokes et al., 2020). There are many possible avenues for combining these approaches with our method, e.g., using coupled autoencoders (Yang et al., 2021) to represent each compound in terms of a combination of other compounds *and* an encoding of its molecular structure.

## 3. Problem Statement

We consider a collection of contexts (e.g., cell types), denoted by $\mathcal{C}$, and actions (e.g., compounds), denoted by $\mathcal{A}$. The outcomes of interest (e.g., gene expression signatures) associated with a given

context $c \in \mathcal{C}$ under action $a \in \mathcal{A}$ are denoted by $\mathbf{x}^{ca} \in \mathbb{R}^p$. Collectively, this forms an order-three tensor $\mathcal{X} = [\mathbf{x}^{ca} : c \in \mathcal{C}, a \in \mathcal{A}] \in \mathbb{R}^{|\mathcal{C}| \times |\mathcal{A}| \times p}$.

**Observations.** To model real-world scenarios, we consider the setup where we not only have access to a sparse subset of the entries in $\mathcal{X}$ (e.g., corresponding to a limited number of historical experiments), but also noisy instantiations of those observations. More formally, let $\tilde{\mathbf{x}}^{ca} \in \mathbb{R}^p$ denote a corrupted version of $\mathbf{x}^{ca}$, e.g., $\tilde{\mathbf{x}}^{ca} = \mathbf{x}^{ca} + \mathbf{e}^{ca}$ or $\tilde{\mathbf{x}}^{ca} = \mathbf{x}^{ca} \circ \mathbf{e}^{ca}$, where $\mathbf{e}^{ca}$ denotes measurement noise and $\circ$ is the entry-wise product. Further, let $\Omega \subset \mathcal{C} \times \mathcal{A}$ denote the context-action pairs for which we observe their associated outcomes, i.e., for each $(c, a) \in \Omega$, we observe $\tilde{\mathbf{x}}^{ca}$. We denote our observation set as $\mathcal{O} = \{\tilde{\mathbf{x}}^{ca} : (c, a) \in \Omega\}$.

**Goal.** Our primary interest is in recovering $\mathcal{X}$ from $\mathcal{O}$.

*Notations.* For any $a \in \mathcal{A}$ and $c \in \mathcal{C}$, let $\mathcal{A}(c) = \{a \in \mathcal{A} : (c, a) \in \Omega\}$ and $\mathcal{C}(a) = \{c \in \mathcal{C} : (c, a) \in \Omega\}$ denote the set of actions which are measured for context $c$, and the set of contexts which are measured for action $a$, respectively. These notations are extended to sets of contexts and actions, i.e., $\mathcal{A}(\mathcal{C}) = \cap_{c \in \mathcal{C}} \mathcal{A}(c)$ and $\mathcal{C}(\mathcal{A}) = \cap_{a \in \mathcal{A}} \mathcal{C}(a)$.

## 4. Algorithm

We propose `SI-A`, which is an extension of the SI estimator introduced in Agarwal et al. (2020), to handle more general sparsity patterns. Though our estimation strategy is applicable for any context-action pair of interest, we consider (without loss of generality) some $(c, a) \notin \Omega$. Below, we introduce additional notations to describe our estimator.

*Additional Notations.* We call $\mathcal{A}(c)$ the "donor actions" for context $c$. Let $\mathcal{C}_{\text{train}} = \mathcal{C}(\mathcal{A}(c) \cup \{a\})$ denote the set of contexts for which all donor actions and the target action are observed, we call these the "training contexts". Let $C = |\mathcal{C}_{\text{train}}|$ and $A = \mathcal{A}(c)$, and define $\tilde{\mathbf{x}}_{\text{train,target}} \in \mathbb{R}^{p \cdot C}$, $\tilde{\mathbf{X}}_{\text{train,donor}} \in \mathbb{R}^{pC \times A}$ and $\tilde{\mathbf{X}}_{\text{test,donor}} \in \mathbb{R}^{p \times A}$ as

$$\tilde{\mathbf{x}}_{\text{train,target}} = [\tilde{\mathbf{x}}^{ia}]_{i \in \mathcal{C}_{\text{train}}} \qquad \tilde{\mathbf{X}}_{\text{train,donor}} = [\tilde{\mathbf{x}}^{ij}]_{i \in \mathcal{C}_{\text{train}}, j \in \mathcal{A}(c)} \qquad \tilde{\mathbf{X}}_{\text{test,donor}} = [\tilde{\mathbf{x}}^{cj}]_{j \in \mathcal{A}(c)}$$

Finally, let `ME` : $\mathbb{R}^{n \times p} \to \mathbb{R}^{n \times p}$ denote any matrix estimation method, used to recover a matrix from noisy observations, such as nuclear norm regularization or singular value thresholding.

**Causal imputation for a novel context-action pair.** We define the our estimation method as follows, where $\dagger$ denotes the pseudoinverse. The method is also presented visually in Fig. 1.

1. Model learning via regression

   (a) Define $\widehat{\mathbf{X}}_{\text{train,donor}} = \text{ME}(\tilde{\mathbf{X}}_{\text{train,donor}})$.

   (b) Define $\widehat{\beta}^{ca} = \widehat{\mathbf{X}}_{\text{train,donor}}^{\dagger} \tilde{\mathbf{x}}_{\text{train,target}} \in \mathbb{R}^{|\mathcal{A}(c)|}$

2. Causal imputation

   (a) Define $\widehat{\mathbf{X}}_{\text{test,donor}} = \text{ME}(\tilde{\mathbf{X}}_{\text{test,donor}})$.

   (b) Define $\widehat{\mathbf{x}}^{ca} = \widehat{\mathbf{X}}_{\text{test,donor}} \widehat{\beta}^{ca}$.

In Appendix A, we discuss how our estimator relates to the original SI estimator.

**Extensions.** We now discuss two natural modifications to the proposed method.

*Reversing the axis of regression.* The method we described learns weights $\widehat{\beta}^{ca}$ which express the action $a$ as a linear combination of other actions $\mathcal{A}(c)$. The same method can be applied to instead express the context $c$ as a linear combination of other contexts, by making the obvious replacements.

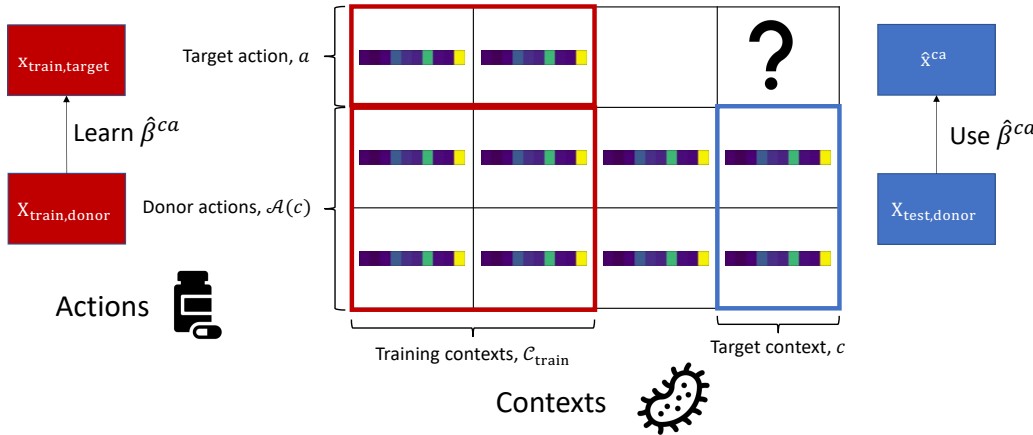

Figure 1: **The SI-Action Method.** To form an estimate $\hat{x}^{ca}$ of the outcome of target action $a$ in target context $c$, we find a set of *donor actions* $\mathcal{A}(c)$ which are available in context $c$. Then, we find *training contexts* $\mathcal{C}_{\text{train}}$ for which both the donor actions and the target action are available, and learn weights $\hat{\beta}^{ca}$ via linear regression. Finally, we use these weights to form $\hat{x}^{ca}$.

*Using both estimators in tandem.* The two estimation strategies described above—weighting actions and weighting contexts—can be used in a complementary fashion. For instance, say we apply the weighting actions estimator first. The imputed values increase the set of donor actions, which should lead to better outcomes when applying the weighting contexts estimator. An analogous argument can be made when the weighting contexts estimator is applied first. This suggests an algorithm that iteratively applies each approach until the imputed tensor converges to a desired threshold. However, one drawback of this approach is that it is computationally more demanding. Formally motivating and analyzing such an algorithm remains to be interesting future work.

*Arbitrary sets of donor actions.* For simplicity, we have written our estimator using *all* actions $\mathcal{A}(c)$ that are available for context $c$ as "donor" actions. The corresponding set of training contexts are those for which each of these actions, along with the target action, is measured, i.e., $\mathcal{C}_{\text{train}} = \mathcal{C}(\mathcal{A}(c) \cup \{a\})$. Expressing $\hat{\beta}$ as a linear combination of all actions $\mathcal{A}(c)$ may be suboptimal if it leads to a significantly smaller set $\mathcal{C}_{\text{train}}$. For example, say $|\mathcal{A}(c)| = 11$, and there is only one training context containing all of these actions, i.e., $|\mathcal{C}(\mathcal{A}(c) \cup \{a\})| = 1$. Suppose there is a subset $\mathcal{A}' \subset \mathcal{A}(c)$, with $|\mathcal{A}'| = 10$ with $|\mathcal{C}(\mathcal{A}' \cup \{a\})| = 100$, i.e., by excluding a single action from the linear combination, we can train on 100 times more contexts. Thus, we may consider the SI estimator induced by a specific *donor set* $\mathcal{A}_{\text{donor}} \subseteq \mathcal{A}(c)$, with the corresponding set of training contexts being $\mathcal{C}_{\text{train}} = \mathcal{C}(\mathcal{A}_{\text{donor}} \cup \{a\})$. The choice of donor set introduces a tradeoff: as the number of donor actions increases, the number of training contexts might decrease. This raises the question of how to pick an optimal set of donor actions for a given prediction. We provide a principled approach to this problem in Section 5.

## 5. Theoretical Results

In this section, we justify our algorithmic approach under a factor model and provide a justification for the factor model through the lens of causal structural models. We focus on the noiseless setting, i.e., for all $(c, a) \in \Omega$, we observe $\tilde{\mathbf{x}}^{ca} = \mathbf{x}^{ca}$, and thus, we may write $\mathbf{X}_{\text{train,donor}} = \hat{\mathbf{X}}_{\text{train,donor}} =$

$\tilde{\mathbf{X}}_{\text{train,donor}}$ and $\mathbf{X}_{\text{test,donor}} = \widehat{\mathbf{X}}_{\text{test,donor}} = \tilde{\mathbf{X}}_{\text{test,donor}}$. Hence, our results should be viewed as one of "identification" as we prove that $\mathcal{X}$ can be identified from noiseless data via our proposed algorithm. Indeed, identification arguments are considered a critical first step in causal inference (Shpitser and Pearl, 2008; Lee et al., 2020a), as they are necessary for any analysis of consistency in the noisy setting. Two noisy settings are of special interest. First, if only a single observation for each $(c, a)$ pair is available, then there is a rich body of literature that establishes formal guarantees for recovering $\mathbf{x}^{ca}$ for all $(c, a) \in \Omega$ using a suite of matrix estimation methods (see Chatterjee (2015) and references therein). Second, if multiple observations are available for each $(c, a)$ pair, then we may average these observations, as we do in Section 6. Under appropriate conditions on the noise $\mathbf{e}^{c,a}$, such as bounded variance in the additive case, this averaging yields a consistent estimator of $\mathbf{x}^{ca}$. Formally building upon these results to provide rigorous sensitivity analyses remains future work.

## 5.1. Identification via `SI-A`

In what follows, we state our key modeling assumptions that will enable our estimator to impute $\mathcal{X}$. To reduce redundancy, the following discussion will be restricted to `SI-A`, i.e., where we learn a regression model across actions. Analogous statements apply to the case of learning across contexts.

First, we recall the definition of a linear factor model, also known as the interactive fixed effects model, which is prevalent within the causal inference literature (Agarwal et al., 2020; Abadie et al., 2010; Bai, 2009; Athey et al., 2021) and critical to the SI framework of Agarwal et al. (2020).

**Definition 1** *We say that $\mathcal{X}$ satisfies a* linear factor model *if, for any $a \in \mathcal{A}$ and $c \in \mathcal{C}$, $\mathbf{x}^{ca} = U^c \mathbf{v}^a$, where $U^c \in \mathbb{R}^{p \times r}$ and $\mathbf{v}^a$ are latent factors associated with the context $c$ and action $a$, respectively.*

*Interpretation.* Definition 1 posits that $\mathcal{X}$ satisfies a low-rank assumption (Kolda and Bader, 2009)[1]. While *exact* adherence to a linear factor model can be a strong assumption, it has been widely observed in practice that many big-data matrices are approximately low-rank. This phenomenon that has recently been motivated by generic latent-variable models (Udell and Townsend, 2019) and theoretically established in several works (Chatterjee, 2015; Agarwal et al., 2019). Since big data matrices with underlying structure are common in biological applications (due, for example, to similarities between cell types and drugs, and interdependence of gene expression values), these works suggest linear factor models as a natural starting point for developing principled causal imputation methods. Further, low-rank approximation can often be empirically verified by inspecting the spectrum of the observations, as we do in Section 6 (see Figure 5a).

**Assumption 1** *Given a target context $c$ and target action $a$, there exists $\beta \in \mathbb{R}^{|\mathcal{A}(c)|}$ such that $\mathbf{v}^a = \sum_{j \in \mathcal{A}(c)} \beta_j \mathbf{v}^j$.*

*Interpretation.* Under the linear factor model, this is a mild assumption, particularly in applications such as genomics, where outcomes are measured for many different actions and contexts. By definition, such a $\beta$ exists if $\mathbf{v}^a$ and $[\mathbf{v}^j]_{j \in \mathcal{A}(c)}$ are linearly dependent. Recall $\mathbf{v}^a$ and $\mathbf{v}^j$ are in $\mathbb{R}^r$. If $r \ll |\mathcal{A}(c)|$, then by the definition of rank, it is easy to see that the 'pathological' case where $\mathbf{v}^a$ is linearly independent of $\{\mathbf{v}^j : j \in \mathcal{A}(c)\}$ is unlikely to hold; that is, in the worst case, this undesirable event occurs for at most $r$ actions out of all possible $\mathcal{A}$, which is a small fraction if $r \ll |\mathcal{A}(c)| \leq |\mathcal{A}|$.

---

[1] Formally, $\mathcal{X} = U \times_2 V$ for some $U \in \mathbb{R}^{|\mathcal{C}| \times r \times p}, V \in \mathbb{R}^{|\mathcal{A}| \times r}$, where $\times_2$ denotes the *mode-2* product.

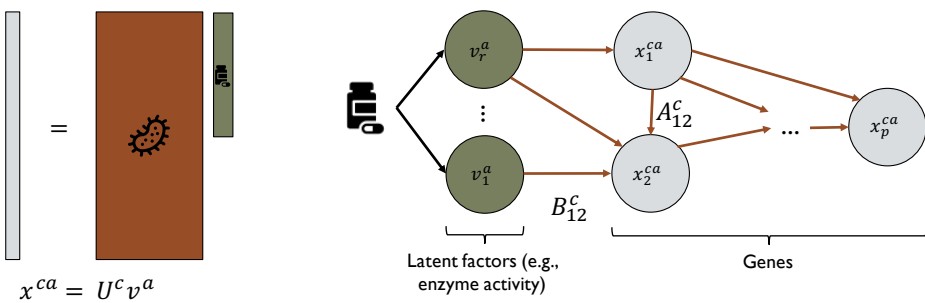

$$x^{ca} = U^c v^a$$

Figure 2: The assumed linear factor model and a linear structural equation model giving rise to it.

**Assumption 2** *Given a target context $c$ and target action $a$, let* $\mathrm{rowspan}(\mathbf{X}_{test,donor})$ *be a subset of* $\mathrm{rowspan}(\mathbf{X}_{train,donor})$,[2] *where $\mathbf{X}_{test,donor}$ and $\mathbf{X}_{train,donor}$ are defined in Section 4.*

*Interpretation.* This assumption states that the target context is no more "complex" than the training context in a linear algebraic sense. This is the key assumption that enables `SI-A` (and thus the original SI estimator) to generalize from a small set of experiments to novel context-actions pairs.

**Theorem 1** *Suppose $\mathcal{X}$ satisfies a linear factor model. Further, for $(c, a) \notin \Omega$, suppose Assumptions 1 and 2 hold. Then $\widehat{\mathbf{x}}^{ca} = \mathbf{x}^{ca}$.*

**Checking identifiability.** Assumption 2 enables SI to recover $\mathcal{X}$ from a sparse set of observations $\Omega$ (Theorem 1). As such, Agarwal et al. (2020) proposed a data-driven hypothesis test to check when this condition is satisfied in practice. Their test statistic $\widehat{\tau} = \|\mathbf{v}_{test} - \mathbf{v}_{train}\mathbf{v}_{train}^T\mathbf{v}_{test}\|_F^2$ measures the gap between the rowspaces of $\mathbf{X}_{train,donor}$ and $\mathbf{X}_{test,donor}$, where $\mathbf{v}_{train}$ and $\mathbf{v}_{test}$ are the right singular vectors of $\mathbf{X}_{train,donor}$ and $\mathbf{X}_{test,donor}$, respectively. Agarwal et al. (2020) derived a critical value $\tau_\alpha$, such that under the null hypothesis $H_0$ where Assumption 2 holds, $\mathbb{P}(\widehat{\tau} \geq \tau_\alpha \mid H_0) \leq \alpha$. However, their critical value depends on the underlying parameters of an underlying noise distribution. Instead of estimating these parameters, we follow a heuristic based on the interpretation of $\widehat{\tau}$ as the spectral energy of $\mathbf{v}_{test}$ that does *not* belong within $\mathrm{span}(\mathbf{v}_{train})$. Specifically, we fix $\rho \in [0, 1]$ and reject $H_0$ if $\widehat{\tau} \geq \rho \cdot \mathrm{rank}(\mathbf{v}_{test})$, i.e., if more than $\rho$ fraction of the spectral energy in $\mathbf{v}_{test}$ lies outside of $\mathrm{span}(\mathbf{v}_{train})$.

Rejection of the test implies that `SI-A` may not perform well on our prediction task, since Assumption 2 is unlikely to hold. In practice, this allows us to switch to a simpler baseline estimator, as we show in Section 6. This hypothesis test also suggests an elegant method for picking a set of donor actions, as discussed in Section 4. For a fixed significance level, we may aim to find a set passing the hypothesis test which *maximizes* the number of training contexts. Unfortunately, this induces a combinatorial optimization problem which may be difficult to solve when there are many actions. We consider two computationally efficient alternatives. First, we may greedily pick actions, according to whichever action *least* reduces the number of training contexts, until the set passes the hypothesis test. Second, we may *always* use $\mathcal{A}(c)$ as the donor set, then use the hypothesis test to decide on whether to use the `SI-A` estimate or a simpler baseline.

---

[2]The row span of a matrix $M \in \mathbb{R}^{m \times n}$ is the linear subspace of $\mathbb{R}^n$ spanned by the row vectors of $M$.

### 5.2. Connecting SI to Structural Causal Models

Here, we provide motivation for a linear factor model in terms of linear structural causal models, which have been frequently used as models of genomic networks (Friedman et al., 2000; Badsha et al., 2019). In particular, we consider the following set of models:

**Definition 2** *A (noiseless)* linear structural equation model *(SEM) over the vector* $\mathbf{z} \in \mathbb{R}^p$ *is defined by the set of equations*

$$\mathbf{z}_i = \sum_{j \in \mathrm{pa}_G(i)} A_{ij}\mathbf{z}_j,$$

*where* $\mathrm{pa}_G(i)$ *denotes the* parents *of node* $i$ *in the directed acyclic graph* $G$.

A particular subclass of linear structural equation models implies a factor model. In particular, we consider the following assumption on the SEM, illustrated in Fig. 2.

**Assumption 3** *Let* $\mathbf{z}^{ca} = (\mathbf{x}^{ca}, \mathbf{v}^a)$, *where* $\mathbf{v}^a$ *depends only on* $a$ *and not on* $c$. *Assume that, for all contexts* $c$, *there exists* $A^c \in \mathbb{R}^{p \times p}$ *and* $B^c \in \mathbb{R}^{p \times r}$ *such that, for all actions* $a$, $\mathbf{z}^{ca}$ *satisfies the following linear structural equation model:*

$$\mathbf{x}^{ca} = A^c \mathbf{x}^{ca} + B^c \mathbf{v}^a \tag{1}$$

*Further assume that and* $\mathbf{v}_i$ *is unobserved for all* $i \in [p]$.

**Proposition 1** *Under Assumption 3,* $\mathcal{X}$ *satisfies a linear factor model.*

**Proof** Under acyclicity of $G$, $A^c$ is lower-triangular up to a permutation, so that $(I - A^c)$ is invertible, and we may re-write $\mathbf{x}^{ca}$ as

$$\mathbf{x}^{ca} = \underbrace{(I - A^c)^{-1} B^c}_{U^c} \mathbf{v}^a$$

Defining $U^c$ as $(I - A^c)^{-1} B^c$ completes the proof. ∎

*Interpretation.* Although Proposition 1 is a simple observation, to the best of our knowledge, this connection has not been previously established. The SI framework of Agarwal et al. (2020), and panel data literature more broadly (Abadie et al., 2010; Arkhangelsky et al., 2019; Athey et al., 2021; Bai, 2009), often imposes a linear factor model structure on $\mathbf{x}^{ca}$. Proposition 1 provides a novel motivation for these models through the lens of structural causal models. In our application, $\mathbf{v}^a$ might represent biological factors, e.g., molecule concentrations or enzyme activity levels, while $U^c$ represents the "gene expression program" (Pope and Medzhitov, 2018) run by cell type $c$.

## 6. Empirical Results

In this section, we perform extensive experimentation on causal imputation for the CMAP dataset.

**CMAP dataset.** Subramanian et al. (2017) developed the L1000 assay, which allows for cost-effective measurement of the gene expression signatures. This assay measures the transcription levels of 978 "landmark" genes, picked via a data-driven approach based on their ability to recover information about the rest of the transcriptome. L1000 signatures have been measured from over

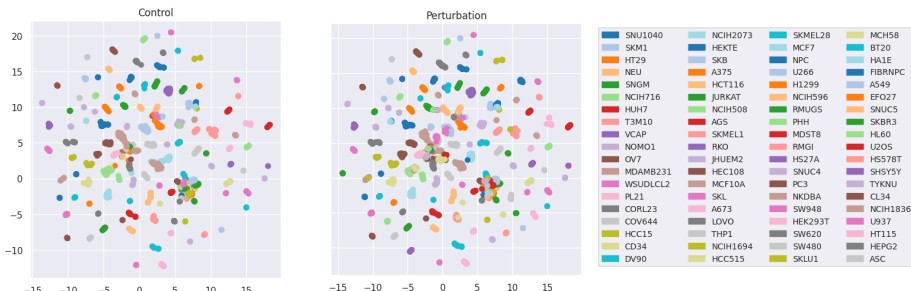

Figure 3: UMAP embedding of gene expression data, colored by cell type.

1,000,000 different samples, covering 71 different cell types and over 20,000 different chemical compounds. We randomly sample 100 of these compounds, along with the "control" compound, *DMSO*, to create a smaller, unbiased version of the dataset. For each cell-type, compound pair, we average all corresponding signatures. This gives a dataset of 519 gene expression signatures. A detailed evaluation of the dataset and a description of our preprocessing pipeline is described in Appendix C. The dataset can be accessed at `https://www.ncbi.nlm.nih.gov/geo/query/acc.cgi?acc=GSE92742`.

**Baseline algorithms.** Figure 3 shows an embedding of 11,185 gene expression vectors via UMAP (McInnes et al., 2018), colored by cell type. Clearly, most of the variation in the data is due to cell type rather than the compound applied to the cell. This is supported by Fig. 8 in Appendix D, where we see that most compound-induced expressions falls within the normal variation of the cell type; the additional variation due to compounds is minor. This suggests a natural *mean-over-actions* baseline

$$\widehat{\mathbf{x}}^{ca}_{\text{avg-a}} = \frac{1}{|\mathcal{A}(c)|} \sum_{a' \in \mathcal{A}(c)} \mathbf{x}^{ca'}. \tag{2}$$

Indeed, it is well-known that the effect of chemical compounds is much smaller than the effect of cell type, so that the simple mean-over-actions estimator already performs well, as we examine shortly. We consider three other natural baselines (see equations in Appendix E). The *mean-over-contexts* is analogous to the mean-over-actions estimator, with the average taken over different contexts from the same action. We may combine these estimators into the parametric *two-way mean* estimator with parameter $\lambda_c \in [0, 1]$, by taking a convex combination of the *mean-over-actions* and *mean-over-contexts* estimators. We use $\lambda_c = 0.5$. The *fixed action effect estimator*, discussed in Section 2, is defined by computing the average shift induced by action $a$ compared to the control, and adding that shift to the control for $c$. Finally, we compare against a variant of SI-A that re-scales contexts rather than actions, as discussed in Section 4, which we refer to as SI-C. Note that standard imputation methods are not scalable to the CMAP dataset: the MissForest implementation in `missingpy` takes 2.5 hours per prediction, and would take $2.5 \times 519/24 \approx 54$ days to run on the subsampled data. `IterativeImputer`, a version of MICE in `sklearn`, is somewhat more scalable, but still prohibitively slow on our subsampled data—see Appendix F.1 for a comparison on 120 signatures.

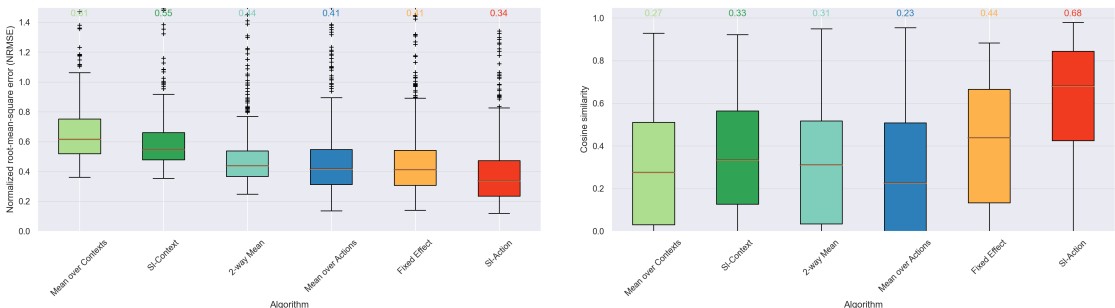

Figure 4: `SI-A` outperforms baselines on causal imputation in the CMAP dataset in terms of NRMSE (**left**) and cosine similarity (**right**) between the true and predicted effects of drugs in each cell type.

## 6.1. Prediction Error

For each algorithm, we measure performance using a leave-one-out (LOO) procedure. In particular, for each cell type and compound pair $(c, a) \in \Omega$ that is measured, we remove the true gene expression signature $\mathbf{x}^{ca}$ from the dataset, and use the remainder of the dataset to estimate $\widehat{\mathbf{x}}^{ca}$. We measure the accuracy of each estimate $\widehat{\mathbf{x}}^{ca}$ in two ways. Normalized root-mean-square error (NRMSE), $\|(\widehat{\mathbf{x}}^{ca} - \mathbf{x}^{ca})/p\|_2 / \text{IQR}(\mathbf{x}^{ca})$, reported in Fig. 4, measures the absolute accuracy to predict the missing gene expression vector[3]. We show NRMSE since it is insensitive to scaling and shifting, but the relative ordering between methods is insensitive to the metric, as seen in Appendix F.2. Meanwhile, cosine similarity between the predicted shift $\widehat{\mathbf{x}}^{ca} - \mathbf{x}^{ca_0}$ and the true shift $\mathbf{x}^{ca} - \mathbf{x}^{ca_0}$ (where $a_0$ is the control action), reported in Fig. 4, measures the degree of *alignment* between the true effect of drug $a$ in cell type $c$ and the predicted effect. Appendix F.3 shows that `SI-A` and the baselines are roughly similar in terms of computation time.

The results of each estimator provide insight into the relationship between cell types and compounds. As expected from Figure 3, the mean-over-actions estimator is a strong baseline, with a median NRMSE of 0.41. In contrast, the mean-over-contexts estimator performs quite poorly. Seeing Figure 3, this is expected: the substantial variation *between* different cell types suggests that each cell type is *not* representable as a linear combination of the others. The fixed effect estimator, using $a' = \text{DMSO}$, performs similarly to the mean-over-actions estimator, indicating its prediction quality is dominated by the cell-type-specific baseline $\mathbf{x}^{ca'}$, i.e., the shift vector for the compound is small. Finally, `SI-A` performs best, with a median NRMSE of 0.34, a 17% reduction compared to the best baseline. The findings on cosine similarity corroborate these results: the drug effects predicted by `SI-A` are significantly more aligned with the true drug effects compared to the baselines.

`SI-A` takes a weighted linear combination of signatures for a given cell type (across different compounds), in contrast to *mean-over-actions* baseline which simply takes equal weights across these signatures. It is noteworthy that such weights are learnt from signatures across compounds for *different* cell types, and yet it is still effective to *transfer* the learnt model to other cell types. This gives credence to the factor model laid out in Section 5.2 holding in our setting. In contrast, `SI-C` performs almost as poorly as the mean-over-contexts baseline, indicating again that the difference between cell types is too great to express one cell type as a linear combination of others.

---

[3] $\text{IQR}(\mathbf{x}^{ca})$ denotes the *interquartile range* for the vector $\mathbf{x}^{ca}$.

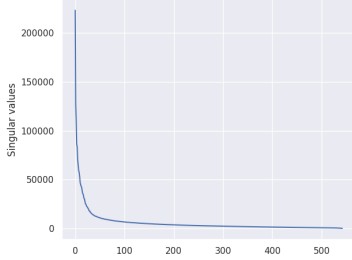

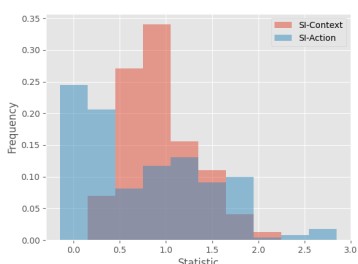

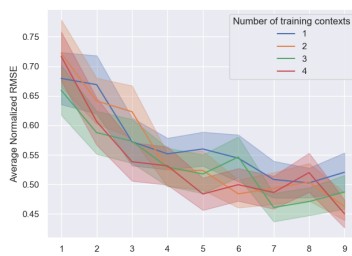

(a) Spectrum of subsampled dataset. 95% of spectral energy is captured in the top 53 singular values.

(b) Distribution of $\hat{\tau}$ when regressing along cell type (context) and compound (action) dimensions.

(c) NRMSE as a function of donor size and training contexts. Error bars show 0.1 standard deviations.

Figure 5: Explaining efficacy of `SI-A` for CMAP dataset.

Notably, we highlight that we do *not* de-noise the data in `SI-A` or `SI-C` to achieve their optimal empirical results, i.e., the results in Figure 4 do not use any `ME` algorithm to remove high amounts of noise that might be present. De-noising is not needed in our setting as the data is implicitly de-noised, since each measurement is an average of multiple observations for a given measured cell type and compound pair $(c, a) \in \Omega$ in the data. Indeed, as shown in Appendix G, applying `ME` does improve prediction if we restrict ourselves to use only a single sample per observed $(c, a) \in \Omega$.

**`SI-A` is effective in biological settings.** As discussed in Section 5, a visual inspection of the spectrum of our observations helps verify the validity of the linear factor model and (by extension) Assumption 1. Fig. 5a shows the singular values of the $519 \times 978$ matrix of gene expression signatures, where over 95% of the spectrum is captured by the top 53 singular values, i.e., $r \approx 53$; hence, the CMAP dataset exhibits the desired low-rank factor structure. To verify Assumption 2, we plot frequencies of the test statistic $\hat{\tau}$ (detailed in Section 4) in Figure 5b. A higher value of $\hat{\tau}$ indicates that Assumption 2 is less likely to hold, and hence the results of our method are less meaningful. We see that the test statistic tends to be lower when regressing in the action dimension, explaining the superior performance of `SI-A` as compared to `SI-C`. This suggests that `SI-A` may be a useful tool in other biological settings, where similar invariance patterns and data availability are common.

**`SI-A` takes advantage of data structure.** Finally, to understand how the performance of `SI-A` depends on the number of donor actions (i.e., $|\mathcal{A}(c)|$) and training contexts (i.e., $|\mathcal{C}_{\text{train}}|$), we restrict our attention to context-action pairs with at least 10 donor actions and at least 5 training contexts, resulting in 75 pairs. For each context-action pair $(c, a)$, and each $(i, j) \in \{1, \ldots, 10\} \times \{1, \ldots, 5\}$, we perform `SI-A` with $i$ random donor actions and $j$ random training contexts. In Fig. 5c, we show the average NRMSE of the predictions over all 75 pairs, for each tuple $(i, j)$. As expected, increasing the number of donor actions leads to a large increase in performance, and increasing the number of training contexts also leads to a large increase in performance, especially when there are few donor actions. The original SI estimator always has $|\mathcal{C}_{\text{train}}| = 1$, but our results highlight the value of using *all* available data to learn the regression model, as is done by `SI-A`.

## 7. Discussion

In this paper, we introduced the `SI-A` estimator, an extension of the SI estimator of Agarwal et al. (2020), for use on a task we call *causal imputation*: predicting the effect of an action across different

contexts. We showed that the `SI-A` estimator provides valid estimates of unseen outcomes under a linear factor model, which we motivate via a connection to structural causal models. We demonstrated the superior performance of `SI-A` to other baselines on the task of causal imputation in the CMAP dataset, an important source of information for predicting the effect of various compounds on gene expression.

Several important directions are left open for future work, of which we cover only a few. First, the tradeoff between the number of donor actions and the number of training contexts raises the need for a principled method for picking "optimal" donor sets. One promising approach may be to frame this choice as a combinatorial optimization problem, where the objective function may be submodular under some assumptions on the problem structure. A related question is whether we can apply SI in a *sequential manner* to infer which samples are most informative to reduce sample complexity in an experimental design and/or active learning framework.

Another important direction for future work is on *nonlinear* methods to the causal imputation problem. Genomic data is known to exhibit highly nonlinear relationships, so that our model in Section 5.2 is only a coarse approximation. A straightforward nonlinear extension of our method would be to perform `SI-A` in a latent space learned by an autoencoder. Two concepts from this paper are likely to be useful in the development and analysis of nonlinear methods. First, we demonstrated that it is beneficial to develop representations for each action which are invariant to the context in which they occur, allowing for the effect of the action to be *transported* between contexts. Second, our mechanistic explanation in Section 5.2 for the success of `SI-A` may serve as a starting point for explaining the success of nonlinear methods.

## Acknowledgments

Chandler Squires was partially supported by an NSF Grad- uate Fellowship, MIT J-Clinic for Machine Learning and Health, and IBM. Caroline Uhler was partially supported by NSF (DMS-1651995), ONR (N00014-17-1-2147 and N00014-18-1-2765), and a Simons Investigator Award. Dennis Shen was partially supported by a Draper Fellowship. Anish Agarwal was partially supported by a MIT IDSS Thomson Reuters Fellowship.

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

# Supplementary Material

## Appendix A.  Comparison with Agarwal et al. (2020)

Agarwal et al. (2020) consider the setting where one observes data associated with a collection of units or contexts (e.g., cell types). Here, there is a pre-intervention period where all units are observed under control (e.g., absence of any actions), followed by a post-intervention period where each unit experiences *one* intervention or action (e.g., compound). Under this particular sparsity pattern, the goal is to impute what would have occurred to *each* unit under *every* intervention in the post-intervention period. In contrast, our work considers a more general observation pattern that allows each unit to potentially experience *multiple* interventions, which is typically present in biological applications. Motivated by the typical data structures found in these domains, we extend the algorithm to regress along interventions (compared to units as is originally proposed), and provide suitable conditions under which the desired counterfactual outcomes are accurately recovered.

## Appendix B.  Proof of Theorem 1

**Proof** Consider any $(c, a) \notin \Omega$. Then by the linear factor model assumption and Assumption 1, for all $i \in \mathcal{C}$

$$\mathbf{x}^{ia} = U^i \mathbf{v}^a \tag{S.3}$$

$$= U^i \Big( \sum_{j \in \mathcal{A}(c)} \beta_j \mathbf{v}^j \Big) \tag{S.4}$$

$$= \sum_{j \in \mathcal{A}(c)} \beta_j (U^i \mathbf{v}^j) \tag{S.5}$$

$$= \sum_{j \in \mathcal{A}(c)} \beta_j \mathbf{x}^{ij}. \tag{S.6}$$

Since (S.6) holds for all $i \in \mathcal{C}$, it necessarily holds for the target context $c$ and those contexts $i \in \mathcal{C}_{\text{train}} = \mathcal{C}(\mathcal{A}(c) \cup \{a\})$.

Now, let $\mathbf{v}_{\text{train}} \in \mathbb{R}^{|\mathcal{A}(c)| \times r_1}$ and $\mathbf{v}_{\text{test}} \in \mathbb{R}^{|\mathcal{A}(c)| \times r_2}$ denote the right singular vectors of $\mathbf{X}_{\text{train,donor}}$ and $\mathbf{X}_{\text{test,donor}}$, respectively, with $r_1 = \text{rank}(\mathbf{X}_{\text{train,donor}})$ and $r_2 = \text{rank}(\mathbf{X}_{\text{test,donor}})$. Recall that $\widehat{\beta} = \mathbf{X}_{\text{train,donor}}^{\dagger} \mathbf{y}_{\text{train}}$. Since $\widehat{\beta} \in \text{rowspan}(\mathbf{X}_{\text{train,donor}}) = \text{span}(\mathbf{v}_{\text{train}})$ by design, it follows from (S.6) that $\widehat{\beta} = \mathbf{v}_{\text{train}} \mathbf{v}_{\text{train}}^T \beta$. Also, Assumption 2 implies that $\mathbf{v}_{\text{train}} \mathbf{v}_{\text{train}}^T \mathbf{v}_{\text{test}} = \mathbf{v}_{\text{test}}$. Combining these arguments yields

$$\widehat{\mathbf{x}}^{ca} = \mathbf{X}_{\text{test,donor}} \widehat{\beta} = \mathbf{X}_{\text{test,donor}} \mathbf{v}_{\text{train}} \mathbf{v}_{\text{train}}^T \beta = \mathbf{X}_{\text{test,donor}} \beta \tag{S.7}$$

$$= \sum_{j \in \mathcal{A}(c)} \beta_j \mathbf{x}^{cj} = \mathbf{x}^{ca}. \tag{S.8}$$

This completes the proof. ∎

## Appendix C. L1000 Dataset

In Figure 6a, we display the availability of gene expression signatures for each cell type/chemical compound pair. The cell types are sorted from left to right by the number of compounds for which gene expression signatures are available. Similarly, the compounds are sorted from bottom to top by the number of cell types for which gene expression signatures are available.

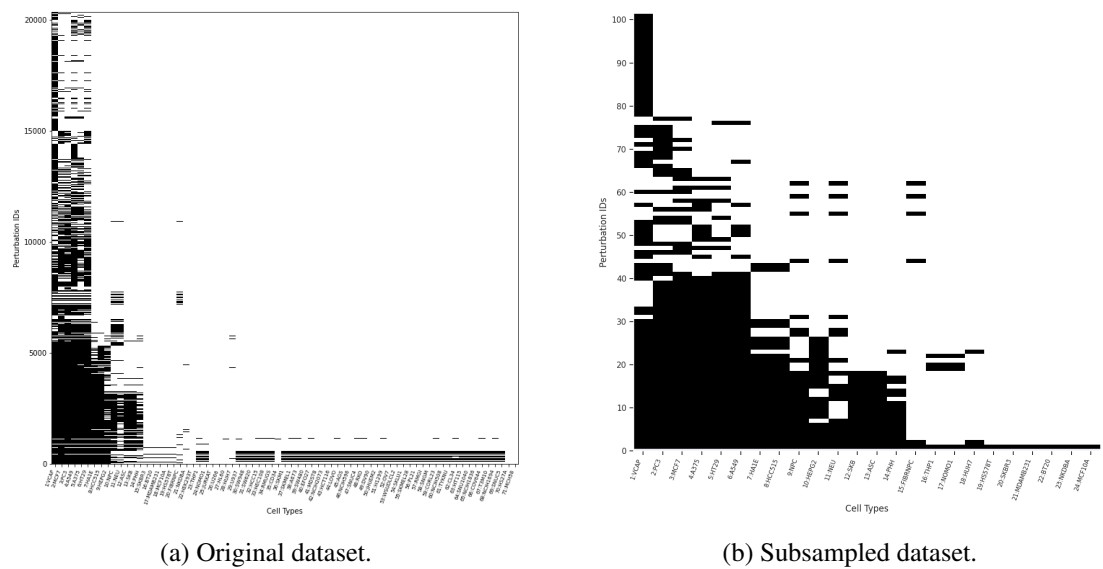

(a) Original dataset.                    (b) Subsampled dataset.

Figure 6: Availability matrix for cell-type compound pairs. A black rectangle indicates that the gene expression profile is available, a white rectangle indicates that it is not.

In Figure 7a, we display the total number of compounds for which gene expression signatures are available, for each cell type. The cell type with the most compounds available is VCAP, with 15,805 compounds available out of 20,369.

In Figure 7b, we display the total number of cell types for which gene expression signatures are available, for each compound. The compound with the most cell types available is DMSO (control), which is available for 70 of 71 cell types, followed by BRD-A19037878, which is available for 64 out of 71 cell types.

**Data Selection.** We use the Level 2 data from L1000 dataset, which contains unnormalized gene expression values. The data is loaded using cmapPy (Enache et al., 2019), available under a 3-clause BSD license. The Level 2 data is split into two sets, "delta" and "epsilon", containing 49,216 and 1,278,882 samples, respectively. They differ in which landmark genes are used; we only use the larger "epsilon" dataset for consistency of our results.

We select 100 compounds at random to run all of our analyses over, in order to create a smaller but unbiased dataset. We show the plots corresponding to those that we showed for the whole dataset in Figures 6b, 7c, and 7d; they qualitatively verify that the subsampled dataset is similar in character to the original.

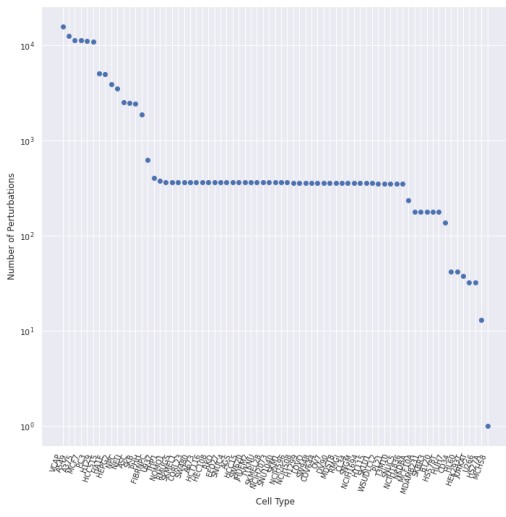

(a) Compounds per cell type in the original dataset.

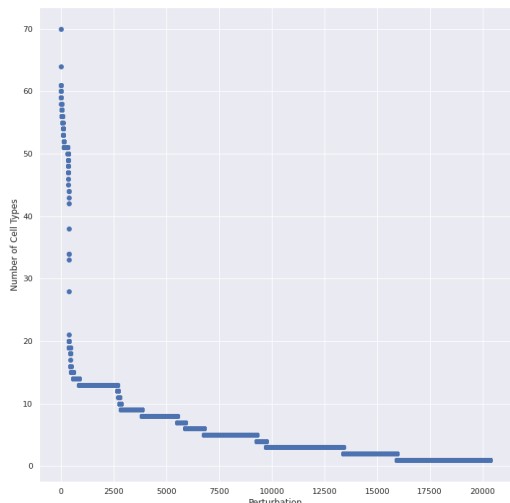

(b) Cell types per compound in the original dataset.

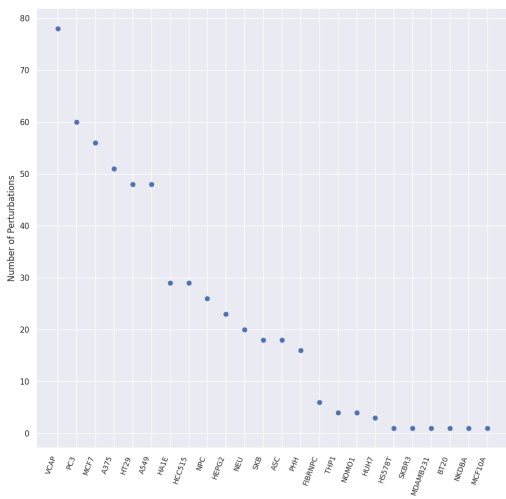

(c) Compounds per cell type in the subsampled dataset.

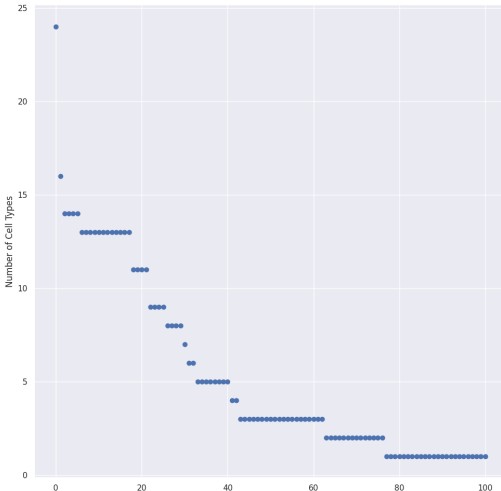

(d) Cell types per compound in the subsampled dataset.

Figure 7: Cross-sectional counts in the original and subsampled datasets.

## Appendix D. UMAP on VCAP data

In Figure 8, we show the UMAP embedding of gene expression data from 70 different compounds in the VCAP cell line, which we picked since it has the greatest number of samples for any cell type in the dataset. Comparing to Figure 3, which included 70 different cell types, we see that gene expression vectors (even within a single cell type) cluster far less by compound than they do by cell type. Moreover, most compounds do not substantially differ from the control (DMSO). This suggests that we should expect estimating the cell-type specific compound effect will be a difficult task.

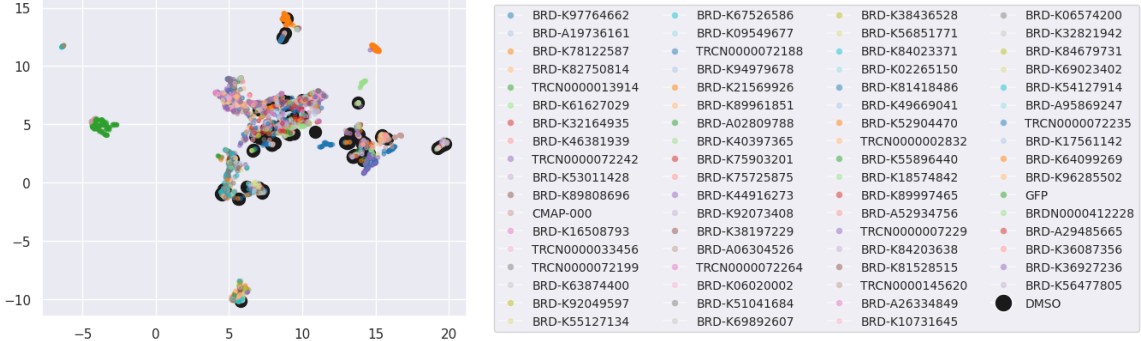

Figure 8: UMAP embedding of gene expression data from the cell type VCAP.

## Appendix E. Baseline Estimators

The *mean-over-contexts* estimator is defined as

$$\widehat{\mathbf{x}}_{\text{avg-c}}^{ca} = \frac{1}{|\mathcal{C}(a)|} \sum_{c' \in \mathcal{C}(a)} \mathbf{x}^{c'a}. \tag{S.9}$$

That is, we average all contexts which receive the target action.

The *two-way mean* estimator with parameter $\lambda_c \in [0, 1]$,

$$\widehat{\mathbf{x}}_{\text{two-way}}^{ca} = \lambda_c \widehat{\mathbf{x}}_{\text{avg-c}}^{ca} + (1 - \lambda_c)\widehat{\mathbf{x}}_{\text{avg-a}}^{ca}, \tag{S.10}$$

is a convex combination of the *mean-over-actions* and *mean-over-contexts* estimators.

Finally, the *fixed action effect estimator*, relative to action $a'$, is defined as

$$\widehat{\mathbf{x}}_{\text{fae}}^{ca} = \mathbf{x}^{ca'} + \widehat{\mathbf{s}}(a' \to a), \tag{S.11}$$

where

$$\widehat{\mathbf{s}}(a' \to a) = \frac{1}{|\mathcal{C}(a)|} \sum_{c \in \mathcal{C}(a)} (\mathbf{x}^{ca} - \mathbf{x}^{ca'}). \tag{S.12}$$

In particular, a natural choice for $a'$ is "control", i.e., no action.

## Appendix F. Additional Empirical Results

### F.1. Results with MICE

In Fig. 9, we reduce the number of randomly picked interventions from 100 to 20, so that MICE can be run in a reasonable amount of time (3 hours). MICE is run with the default parameters for the `IterativeImputer` class in `sklearn` as of May 28th, 2021, including `max_iter=10` and `tol=0.001`. See the `IterativeImputer` documentation for more details.

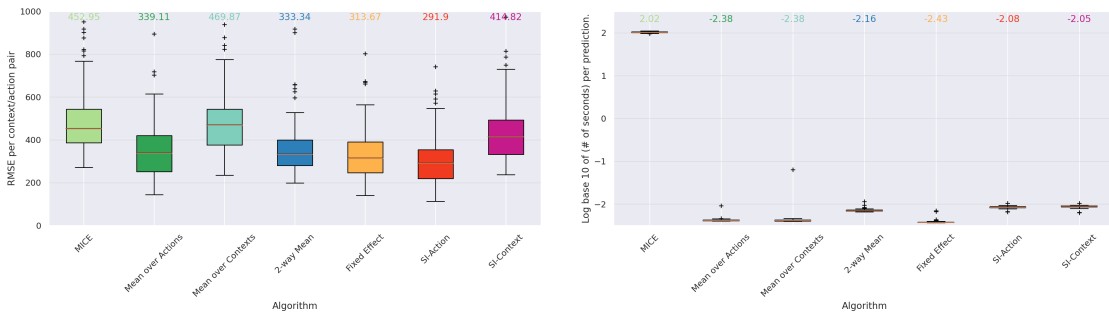

Figure 9: MICE is roughly 4 orders of magnitude slower than `SI-A` and delivers has poor estimation performance compared to `SI-A` and the other baselines.

### F.2. $R^2$

Fig. 10 demonstrates the performance on the leave-one-out (LOO) prediction task in Section 6 on various metrics. The relative ordering of method performance remains the same.

### F.3. Computation Time

Fig. 11 demonstrates the time (on log scale) required for each leave-one-out (LOO) prediction task from Section 6. All methods are highly efficient, taking less than 0.1 seconds on almost all instances and roughly 0.01 seconds on most instances. The baselines are, on typical instances, roughly 2x faster than the methods based on synthetic interventions. The time required by synthetic interventions varies more due to the varying number of donor actions and training contexts.

## Appendix G. Results on Single Samples

Figure 12 shows the results of using `SI-A` on *unaveraged* data. In particular, for each cell type and compound, we select a single corresponding sample at random.

As described in Figure 4, we can apply some `ME` to de-noise our observations prior to regression. To this end, we consider the hard singular value thresholding (HSVT) estimator at energy level $\rho = 0.95$, that is, given the singular value decomposition $X = U\Sigma V^\top$, with singular values in decreasing order, we find that first $k$ such that $\sum_{i=1}^{k} \Sigma_{ii}^2 \geq \rho\|X\|_F^2$, and use $\mathrm{HSVT}(X, k) = \sum_{i=1}^{k} \Sigma_{ii} U_i V_i^\top$.

We see that, as predicted by theory, matrix estimation improves the predictions on unaveraged data (`SI-A` vs. `SI-A`-HSVT). However, the results still do not match the performance of the simple mean-over-actions baseline.

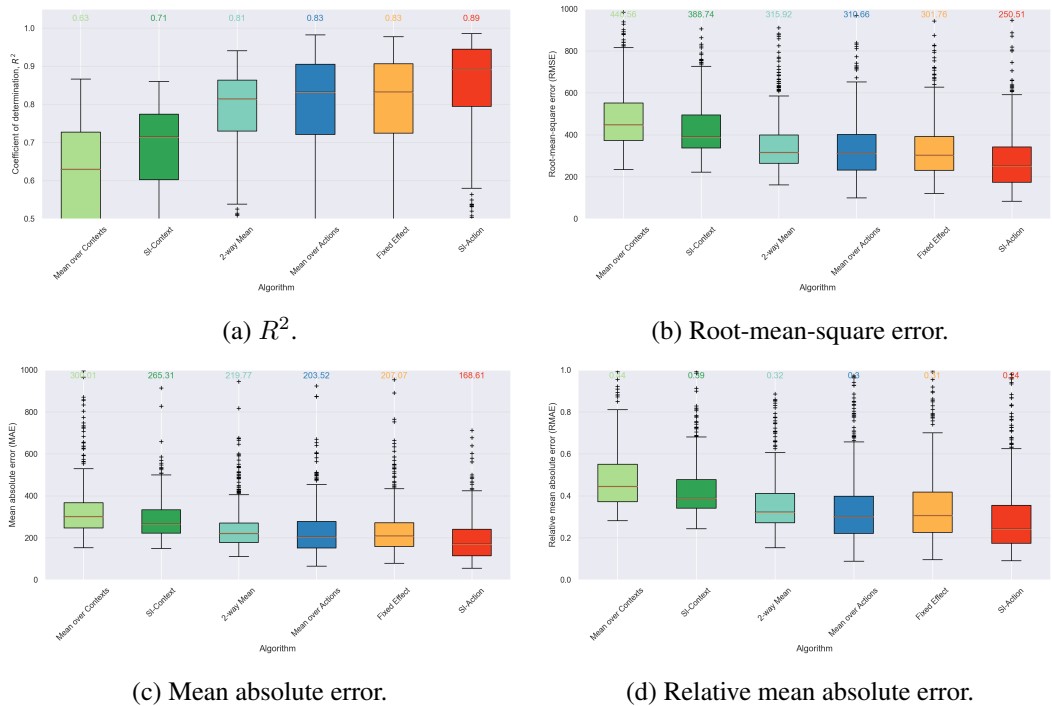

(a) $R^2$.

(b) Root-mean-square error.

(c) Mean absolute error.

(d) Relative mean absolute error.

Figure 10: The relative rankings of SI-A and baselines in terms of $R^2$ are similar across different metrics, with SI-A performing best, followed by mean-over-actions and the fixed effect estimators.

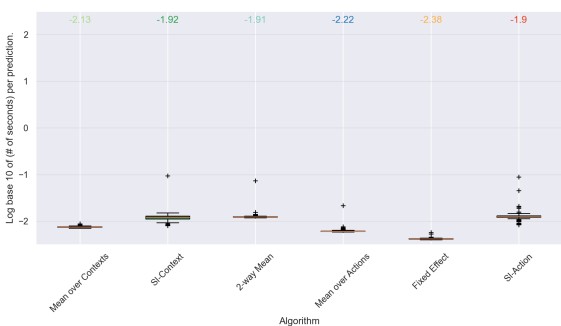

Figure 11: Computation Times of SI-A and baselines. All methods are highly scalable, and SI-A is only about 2x slower than the simple baselines.

Thus, prior to prediction, we perform the subspace inclusion hypothesis test described in Section 4. In particular, if $\widehat{\tau} \geq 0.1 \cdot \text{rank}(\mathbf{X}_{\text{test}})$, then we reject the hypothesis test, concluding that the SI method is unlikely to work. If the test passes, we use the SI-A-HSVT predictor; if it fails, we instead use the mean-across-actions predictor as a strong "fallback" option.

We see that adding this hypothesis test (SI-A-HSVT, +test) returns us to the performance level of the mean-across-actions baseline.

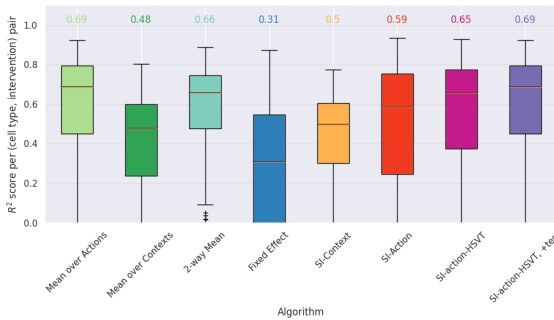

Figure 12: Performance of causal imputation algorithms on recovering the effects of compounds in the CMAP dataset, using *unaveraged* data.

