# OpenReview forum: "Causal Imputation via Synthetic Interventions"
_cclear.cc/CLeaR/2022/Conference — CLeaR 2022 Poster_

### Official Review · Reviewer_mRBv · 2021-11-08

**Confidence:** 3
**Overall Score:** 8

**Main Review:**

#### __Conclusion__
The solution proposed in this paper is to the best of my knowledge original. The paper outlines a scaleable approach for imputing datasets, and although the paper would be strengthened by a theoretical analysis of the behaviour in the presence of noise,  the contributions of the paper are significant. In addition to this, the paper is well-written and well-motivated, and for those reasons, I believe it would be suited for publication at CLeaR.

#### __Strengths__
1. Clarity. The paper is well written and presented, and is the method is well illustrated by the motivating example.
2. Application. The paper proposes a framework that addresses a very concrete problem, and demonstrates real-world applicability by superior performance in the CMAP gene data set.

#### __Weaknesses__
3. Weak theoretical results. Under the latent-variable model and rank assumptions, the identifiability result is hardly surprising for the noiseless case. The paper would have been strengthened by an analysis of the noisy case.

#### __Minor comments__
4. I do not fully understand the ‘selection diagram’ that are e.g. used in Table 1. Could you be more elaborate on how this relates to a causal model?
5. In `Additional Notations’ on p. 5, you write $\tilde X_{\text{train,donor}}$ twice. I believe the first should be $\tilde X_{\text{train,target}}$. Also in Assumption 2, you state that $x_{\text{train}}, x_{\text{test}}$ are defined in Section 4, but they are not.
6. In Assumption 3, one could misunderstand that this is the SCM for only for a particular action $a$. Could you be more elaborate on the assumption that $x^{ca}$ follows this SCM for all actions $a$ (so in particular $A^c, B^c$ are shared across all those SCMs).
7. Errors in the references: Agarwal (2020a, 2020b) need references to ArXiV and NeurIPS. And Chatterjee (2015a, 2015b) are the same reference.

**Summary:**

The authors propose causal imputation, a method to infer the effect of actions in various contexts, when only a subset of all context-action-pairs are observed. The method is motivated by prediction of the effect of chemical compounds (actions) in cells (contexts).  Prediction is performed by learning regression coefficients in a set of training contexts, where the same `donor actions’ are observed as in the target context c. Assuming that there is no noise, and that the data obeys a latent factor representation, the authors show identifiability under rank assumptions. Additionally, they connect this latent factor model to structural equation models.  The paper evaluates performance in a gene expression data set, and obtains better performance than competing methods.

---

> ### Author Response · Authors · 2021-12-03
> **Response to mRBv**
>
> Thank you for your positive remarks! We are glad that you found the writing and presentation clear and the method applicable. We agree that further analysis of the noisy case would make the paper even stronger, but we believe that our method, our initial identifiability result, and our extensive application to the CMAP dataset constitute significant contributions.
>
> ## Minor comments
>
> 4. Please refer to e.g. “Generalized Transportability: Synthesis of Experiments from Heterogeneous Domains” (Lee et al., 2020) for a recent exposition on selection diagrams. Briefly, selection diagrams are simply causal models which record how mechanisms (i.e., conditional distributions of variables given their parents) change between different contexts, by including additional nodes to represent the contexts.
>
> 5. Thank you for the catch, you are right, the first occurrence of $\tilde{X}_{train,donor}$ should be changed, and the quantities in Assumption 2 should have the additional subscript “donor”.
>
> 6. Thanks for pointing out this potential source of confusion regarding Assumption 3. We will clarify that there exists $A^c, B^c$ for each cell type $c$ and $v^a$ for each action $a$ such that for all $(c, a)$, $x^{ca} = A^c x^{ca} + B^c v^a$
>
> 7. We will fix these references.

---

> > ### Comment · Reviewer_mRBv · 2021-12-12
> > **Response to the authors**
> >
> > I thank the authors for their response and their clarifications.
> > Regarding 4., it would be great if the authors could add this brief elaboration to the paper.

---

### Official Review · Reviewer_LHjH · 2021-11-21

**Confidence:** 3
**Overall Score:** 5

**Main Review:**

This paper studies a specific identification task where the unobserved variable is a set of potential outcomes $X^{c,a}$ where C represents a context and A means an action. For each instance C = c, A = a, only a corrupt reading $\tilde{X}^{c, a} = X^{c, a} + e^{c, a}$ where  $e^{c, a}$ is an arbitrary measurement noise. The goal is to recover the actual $X^{c, a}$ from the corrupt observations $\tilde{X}^{c, a}$. The authors propose an algorithm to solve this task.

Overall, I found this paper quite confusing. Several notations and technical concepts do not seem to be well-defined. I will summarize my questions below.

>"Observations set as $\mathcal{O} = (\tilde{x}^{c, a}: (c, a) \in \Omega )$"

Do the authors imply that the learner could only obtain a single observation for each context-acton pair (c, a)? If this is the case, it does not seem feasible to recover the actual value $x^{c, a}$ if the measurement noise $e^{c, a}\neq 0$.

> "$e^{ca}$ denotes measurement noise"

Suppose the leaner could obtain multiple observations for every $c, a$. The problem setting still seems ill-posed. For instance, let $P(e^{c, a} = 0) = P(e^{c, a} = 1) = 0.5$ and $\tilde{x}^{c, a} = x^{c, a} \oplus e^{c, a}$. In such setting, it is unclear if there exists any method to recover the actual $x^{c, a}$. Do the authors assume additional assumptions about the noise $e^{c, a}$, e.g., it follows a normal distirbution? Please elaborate.

> "Definition 2 A (noiseless) linear structural causal model ..."

This brings me to Definition 2 where the authors seem to derive the identifiability result based on a fully deterministic linear SCM. That is, the measure noise $e^{c, a} = 0$. Am I missing some nuances here? If this is true, this identification condition (noiseless) is too restrictive to be practical. It is hard to imagine a real-life system where uncertainty does not exist.

> "Assumption 3"

The linear equation in Eq. (1) seems curious. It reads
$$
x^{ca} = A^c x^{ca} + B^c v^a.
$$
That is, the underlying SCM is cyclic, while the diagram in Figure 2 seems to suggest it being an acyclic model. At the first, I thought it was a typo, but the proof for Proposition 1 seems to suggest otherwise.

**Summary:**

Review of Paper#108

---

> ### Author Response · Authors · 2021-12-03
> **Response to LHjH**
>
>
> Thanks for the helpful comments!
>
> ## Measurement noise and recovery of $x^{ca}$
>
> You are correct that some requirements are needed on the measurement noise.
>
> Such assumptions could take one of two forms.
>
> (1) If multiple observations $\tilde{x}^{ca}$ are available for each $(c, a)$, then we can average these values to obtain a single vector. Under very weak assumptions (e.g., integrability of $e^{ca}$ in the additive noise setting), these averages are consistent estimates of their mean, $x^{ca}$.
>
> (2) if only a single observation is available for each $(c, a)$ we may use a matrix estimation method for denoising these observations, as we suggested in Section 4. There is an extensive literature on matrix denoising that can provide conditions under which consistent denoising is possible.
>
> We give our identifiability result in the noiseless case to abstract away the particular details of how consistent estimates of $x^{ca}$ for $(c, a) \in \Omega$ are obtained. We will add a remark to make it more clear why we give our identifiability result in the noiseless setting.
>
>
> ## Acyclicity of the model
>
> Equation (1) does not imply that the underlying SCM is cyclic, and the proof of Proposition 1 does not suggest that it is cyclic. In fact, Proposition 1 explicitly uses the fact that the underlying SCM is acyclic, and this is explicitly stated. The form of Equation (1) is a typical way of writing linear structural causal models, see for instance Section 2 of “Direct Estimation of Differences in Causal Graphs” (Wang et al., 2018).

---

### Official Review · Reviewer_JALQ · 2021-11-25

**Confidence:** 2
**Overall Score:** 6

**Main Review:**

Originality and Significance:
Prior work on synthetic interventions estimates the "target" effect by constructing the synthetic version of the target context using a weighted combination of donor contexts. This paper further extends this idea also on the action side, by constructing the target action as a weighted combination of donor actions and propose a new estimator called SI-A, which has strong theoretical property and yields stronger empirical performance.

Clarity:
This paper is well-written and easy to follow.

Quality:
The method is well-illustrated, and it would be great to include some motivations for extending the SI idea on the action part. Theoretically, the method is justified under a factor model. Empirically, it shows stronger performance than the original SI estimator. Some minor comments here: is it possible to characterize/quantify the improvements of SI-A over SI? or under which regimes, SI-A is preferable than SI?

**Summary:**

This paper proposes an extension of the synthetic intervention estimator to handle causal imputation.

---

> ### Author Response · Authors · 2021-12-03
> **Response to JALQ**
>
> Thanks for the positive feedback! It may be difficult to explicitly quantify the improvement of SI-A over SI in general settings, but we would like to emphasize that Figure 5(b) demonstrates a way to empirically discern which method is more fitting.

---

### Decision · Program_Chairs · 2022-01-13

**Decision:**

Accept (Poster)

**Comment:**

The authors propose a method to infer the effect of an action in a context, given a set of observed action-context pairs. In a noiseless factor model, the authors study the identifiability of the model. Additionally, they connect this latent factor model to structural equation models. The reviewers point out that the noiseless setup is unrealistic; understanding the behaviour of methods in the noisy finite-sample case is important. Even though the theoretical setup is not very realistic, overall this is a valuable addition to the literature.